



# A statistical and process oriented evaluation of cloud radiative effects in high resolution global models

Manu Anna Thomas[1], Abhay Devasthale[1], Torben Koenigk[1], Klaus Wyser[1], Malcolm Roberts[2], Christopher Roberts[3], and Katja Lohmann[4]

[1]Swedish Meteorological and Hydrological Institute, Folkborgsvägen 17, 60176 Norrköping, Sweden.
[2]Met Office Hadley Centre, FitzRoy Rd, Exeter, Devon EX1 3PB, United Kingdom.
[3]European Centre for Medium-Range Weather Forecasts | ECMWF, Shinfield Park, Reading RG2 9AX, United Kingdom.
[4]Max Planck Institute for Meteorology, Bundesstr. 53, D-20146 Hamburg, Germany

**Correspondence:** Manu Anna Thomas (Manu.Thomas@smhi.se)

**Abstract.**

This study evaluates the impact of atmospheric horizontal resolution on the representation of cloud radiative effects (CREs) in an ensemble of global climate model simulations following the protocols of the High Resolution Model Intercomparison Project (HighResMIP). We compare results from four European modelling centres, each of which provides data from 'standard' and 'high' resolution model configurations. Simulated radiative fluxes are compared with observation-based estimates derived from the Clouds and Earth's Radiant Energy System (CERES) dataset. Model CRE biases are evaluated using both conventional statistics (e.g. time and spatial averages) and after conditioning on the phase of two modes of internal climate variability, namely the El Niño and Southern Oscillation (ENSO) and the North Atlantic Oscillation (NAO). Simulated top-of-atmosphere (TOA) and surface CREs show large biases over the polar regions, particularly over regions where seasonal sea-ice variability is strongest. Increasing atmospheric resolution does not significantly improve these biases. The spatial structure of the cloud radiative response to ENSO and NAO variability is simulated reasonably well by all model configurations considered in this study. However, it is difficult to identify a systematic impact of atmospheric resolution on the associated CRE errors. Mean absolute CRE errors conditioned on ENSO phase are relatively large (5-10 W/m$^2$) and show differences between models. We suggest this is a consequence of differences in the parameterization of SW radiative transfer and the treatment of cloud optical properties rather than a result of differences in resolution. In contrast, mean absolute CRE errors conditioned on NAO phase are generally smaller (0-2 W/m$^2$) and more similar across models. Although the regional details of CRE biases show some sensitivity to atmospheric resolution within a particular model, it is difficult to identify patterns that hold across all models. This apparent insensitivity to increased atmospheric horizontal resolution indicates that physical parameterizations play a dominant role in determining the behaviour of cloud-radiation feedbacks. However, we note that these results are obtained from atmosphere-only simulations and the impact of changes in atmospheric resolution may be different in the presence of coupled climate feedbacks.



## 1 Introduction

Clouds cover about 70% of the Earth's area and have multiple effects on climate (Karlsson and Devasthale (2018); Stubenrauch et al. (2013)). They regulate the Earth's radiation budget by modulating the incoming solar radiation as well as the outgoing longwave radiation (Stephens et al. (2018)). Cloud processes occur from micrometer (e.g. condensation or freezing)
to kilometer scales (e.g. convective systems). Clouds also have a strong dynamic character and vary substantially in space and time in the atmosphere (Steiner et al. (2018)). Given the complexity of cloud-climate interactions, cloud processes are heavily parameterized in climate models. Considering their tight coupling to the radiation budget, they are one of the key components of the Earth system that need to be evaluated in the global climate models. Evaluating clouds requires a two-pronged approach, wherein both statistical and process-oriented comparisons with observations are needed. In the former, the absolute biases in
cloud properties and cloud radiative effects by statistical comparisons of mean fields is carried out, whereas, the degree with which a certain cloud process is simulated by climate models is assessed in the latter.

Atmospheric processes, especially those related to cloud-climate interactions, are sensitive to the spatial resolution of climate models. For example, increasing the spatial resolution in models is shown to be crucial to accurately reproduce the large scale features such as El Niño Southern Oscillation (Shaevitz et al., 2014; Masson et al., 2012), Inter Tropical Convergence
Zone (ITCZ) (Doi et al., 2012), jet streams (Lu et al., 2015; Sakaguchi et al., 2015), and storm tracks (Hodges et al., 2011). Improvements are also seen in the simulation of synoptic scale phenomena such as tropical cyclones (Murakami et al., 2015; Walsh et al., 2015; Shaevitz et al., 2014) and polar lows (Zappa et al., 2014). A detailed overview of the improvements in the key climate processes is addressed in Haarsma et al. (2016). In light of these studies, the EU funded PRIMAVERA (PRocess-based climate sIMulation: AdVances in high resolution modelling and European climate Risk Assessment) project
(https://www.primavera-h2020.eu/) aims at improving our understanding of the role an increased spatial resolution plays in simulating climate processes and their feedbacks.

Here, in the context of this PRIMAVERA project, the surface and top of the atmosphere cloud radiative effects (CREs) are evaluated in 4 global climate models with varying resolutions adding up to a total of 9 different set-ups using satellite observations from the NASA's CERES-EBAF (Clouds and the Earth's Radiant Energy System-Energy Balanced And Filled)
instrument. CERES provides the longest, continuous space based global observations of cloud forcings. Evaluating climate models provides a positive feedback loop, wherein as the climate models improve, in part due to better observations, the requirements on observations have also increased (Flato et al., 2013; Ferraro et al., 2015; Webb et al., 2017). Particularly, the last decade has seen an exponential increase and maturity in observations and, as a result, has provided greater insights into model deficiencies and limitations (Reichler and Kim, 2001; Tian et al., 2013; Teixeira et al., 2014; Baker and Taylor, 2016).
In the present study, we carry out evaluations using both approaches, i.e. the statistical and process oriented comparisons. For the latter, we focus on two major modes of natural variability, namely El Niño Southern Oscillation (ENSO) and North Atlantic Oscillation (NAO), that govern the atmospheric variability in the tropical Pacific and North Atlantic Oceans and the surrounding continents. First the typical cloud radiative response to ENSO and NAO is investigated, and then we test how well this response is simulated by climate models. Cloud radiative response is defined as the change in cloud radiative effects





observed during the positive and negative phases of ENSO and NAO compared to climatology. We further investigated if high spatial resolution adds value while capturing the cloud radiative response during these two major modes of natural variability.

## 2 Models, observations and methods used in the study

### 2.1 Models participated in the PRIMAVERA project

5  The shortwave (SW), longwave (LW) and combined cloud radiative effects (CREs) are evaluated in the High Resolution Model Intercomparison Project, HighResMIP (Haarsma et al., 2016) models with varying resolutions that participated in the PRIMAVERA project. A brief description of these models used in this study is provided in the table below. All these models are atmosphere-only forced HadlSST2.2 (Kennedy et al., 2017) simulations that include both SST and sea ice concentrations. The HadGEM3 model is the only model that is run at three different spatial resolutions at approximately 40, 90 and 200 kms (at the

10  equator). All the other models are run at two different horizontal resolutions as shown in the Table 1. Longer simulations from 1950-2014 were carried out with these models as part of HighResMIP, however, the period from 1982-2014 is used for this study. Each model uses its own background aerosol climatology. However, the aerosol forcing from the anthropogenic sources is generated by the MACv2-SP method proposed by Stevens et al. (2017). By this method the aerosol forcing is calculated based on the aerosol optical properties and fractional change in cloud droplet number concentrations. More details of these

15  high resolution simulations (HighResMIPv1.0) are given in Haarsma et al. (2016). Monthly means of SW and LW, clear sky and all sky fluxes are used to derive the CREs. The CREs at the top of the atmosphere (TOA) and surface (SFC) is defined as the difference between all sky and clear sky fluxes.

**Table 1.** List of the models analyzed in this study

| Models used | Grid name | Resolution at 0N | Resolution at 50 N | Atmosphere | References |
|---|---|---|---|---|---|
| HadGEM3-GC31-HM | N512L85 | ~40 km | ~25 km | MetUM-GA7.1 | Williams et al. (2017) |
| HadGEM3-GC31-MM | N216L85 | ~90 km | ~60 km | MetUM-GA7.1 | Williams et al. (2017) |
| HadGEM3-GC31-LM | N96L85 | ~200 km | ~130 km | MetUM-GA7.1 | Williams et al. (2017) |
| EC-Earth3-HR | T511L91 | ~40km | ~35km | IFS CY36r4 | Haarsma et al. (2018) |
| EC-Earth3 | T255L91 | ~80km | ~70km | IFS CY36r4 | Haarsma et al. (2018) |
| MPIESM-XR | T255L95 | ~50 km | ~35 km | ECHAM6.3 | Stevens et al. (2013) |
| MPIESM-HR | T127L95 | ~100 km | ~65 km | ECHAM6.3 | Stevens et al. (2013) |
| ECMWF-HR | Tco399L91 | ~25 km | ~25 km | IFS CY43r1 | Roberts et al. (2018) |
| ECMWF-LR | Tco199L91 | ~50 km | ~50 km | IFS CY43r1 | Roberts et al. (2018) |

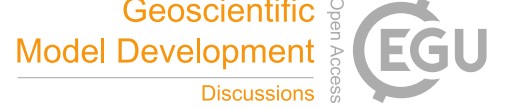

## 2.2 CERES-EBAF

The model simulated TOA and SFC CREs for the Dec-Jan-Feb (DJF) and Jun-Jul-Aug (JJA) averaged months are evaluated against the CERES-EBAF satellite observational data (https://ceres-tool.larc.nasa.gov). The CERES instrument aboard NASA's satellites aims at understanding the clouds and Earth's energy budget. The first CERES instrument was launched

aboard NASA's Tropical Rainfall Measurement Mission (TRMM) in 1997 and thereafter similar instruments were flown onboard three satellite missions, namely Terra and Aqua satellites and Suomi National Polar-orbiting Partnership (S-NPP) satellite. The clear and all sky TOA and SFC fluxes are available at a $1\times1^o$ resolution for the period 2000-2016. For the fluxes at the TOA and SFC, the CERES_EBAF_TOA_Ed4.0 version (Loeb et al., 2009) and CERES_EBAF_Surface_Ed2.8 (Kato et al., 2013) are used respectively. CERES cloud forcing and flux datasets have been used in a number of studies for model evalua-

tions (Wang and Su, 2013, 2015; Stanfield et al., 2015; Calisto et al., 2014). For the analysis, the model data is also re-gridded to a $1\times1^o$ grid. However, in order to increase the number of cases with enhanced positive and negative phases of ENSO and NAO, we consider the whole time period in our simulations from 1982-2014 even though the observational reference period is shorter. In the era of both Terra and Aqua satellites (i.e. from 2002 onwards), both the global and regional uncertainties in the CERES-EBAF TOA and SFC fluxes are reduced dramatically. The typical overall uncertainty, after considering the uncer-

tainties in the calibration, diurnal corrections and radiance-to-flux conversions, in the TOA SW and LW remain in the range of 2-5 W/m$^2$. The uncertainties in the surface fluxes are higher, typically in the range 5-18 W/m$^2$. The detailed data quality summaries are found at the following links:

(https://ceres.larc.nasa.gov/documents/DQ_summaries/CERES_EBAF_Ed4.0_DQS.pdf)

(https://ceres.larc.nasa.gov/documents/DQ_summaries/CERES_EBAF-Surface_Ed4.0_DQS.pdf)

## 2.3 ENSO analysis

ENSO is the leading mode of interannual climate variability in the tropics where it has an impact on the Walker circulation and the local Hadley circulation thereby having a big response in the CREs (Cess et al., 2001b, a). To compute the CRE response to ENSO, first, Niño3.4 index is computed to extract the positive and negative phases of El Niño. This index is based on the sea surface temperature (SST) anomalies over the Niño3.4 region (5N-5S, 170W-120W). When the SST anomalies

over this region are positive (negative) and more (less) than one standard deviation, ENSO is considered to be in a stronger positive (negative) phase (denoted hereafter as ENP and ENN respectively). This method is applied to all the models used in this study to extract the months when these phases are encountered. For our reference data set, CERES, the positive and negative phases are chosen from observations (https://www.esrl.noaa.gov/psd/enso/). The TOA and SFC cloud radiative fluxes associated with these phases are then computed. To extract the cloud response associated solely by the ENP and ENN phases,

the differences from the monthly climatological CREs are taken. This would give the change in the CREs during El Niño/La Niña years with respect to normal years. Instead of evaluating the ensemble mean of the CRE response from all the models, we split the CRE response into an ensemble mean of models with high resolution (Hi-res) and their low/standard (Std-res)





resolution counterparts. The models that are included in the Hi-res ensemble mean are HadGEM3-GC31-HM, EC-Earth3-HR, MPIESM-XR and ECMWF-HR. Their respective low resolution counterparts are included in the Std-res ensemble mean.

## 2.4  NAO analysis

NAO is the most prominent mode of winter variability in the North Atlantic region. To evaluate the CREs associated with the
positive and negative phases of NAO, the standard NAO Index is calculated by taking the difference between normalized sea level pressure (SLP) anomalies between Ponta Delgada, Azores (southernmost point) and Stykkisholmur, Iceland (northern most point) (Stoner et al., 2009). To extract the stronger positive and negative phases of NAO in the observational reference data set, the NAO indices are calculated using the SLP from ERA-Interim data. This study focuses on stronger and weaker than normal NAO phases. If the NAO index is positive (negative) and is more (less) than one standard deviation, the NAO is
considered to be in the stronger positive (negative) phase (NAOP/NAON). For this analysis, we consider the extended winter period from November through April. This method is followed to compute the NAO indices in all the models. As in the case of the ENSO analysis, to quantify the response of the NAO to the TOA and SFC CREs, the difference of the CREs associated with the phases from the climatological mean are taken. Here, since the focus is on the winter half of the year, the seasonal climatological mean is considered. Here, too, the CRE response based on Hi-res and Std-res model set ups are analyzed
separately.

## 3  A statistical evaluation of the cloud radiative effects

In this section, the statistical comparison and evaluation of TOA and SFC cloud radiative effects and their sensitivity to model resolution are presented in Figs.1 to 4. The SW and LW components are evaluated separately for DJF mean (left) and JJA mean (right) seasons and are presented as zonally averaged differences from the observations. Also shown are the net CREs
(i.e. SW+LW). The grey envelopes in Figs.1 and 4 show one standard deviation of CREs in the CERES observations over the 16-year period, as a measure of natural interannual variability in the zonal means.

### 3.1  CREs at the TOA

In DJF, all models, irrespective of their resolution, overestimate the SW TOA CRE by 20 – 40 W/m$^2$ over the bright and persistent decks of Southern Ocean clouds (Fig.1(left)). This overestimation is well above the expected variability seen in the
observations (grey envelope). A clear distinction can be seen in the MPIESM model, where the lower resolution simulation has the lowest positive bias compared to the other models. All the models underestimate the SW TOA CRE by 10 W/m$^2$ over the convective regimes in the southern hemisphere (SH), while HadGEM3 set ups better simulate this response, irrespective of their resolution. Over the tropical belt, the two models, HadGEM3 and MPIESM show a positive bias by up to 15 W/m$^2$ and the other models seem to have a slight negative bias. On the contrary, the LW CREs are underestimated by all the models over
this region. Here too the biases are significantly higher than the observational variability. The low resolution versions of the





respective models better simulates the LW effects in DJF mean over the tropical belt. The high biases in the SW CREs in the south are clearly seen in the combined response.

In JJA months, large discrepancy is seen north of 30N in the SW CREs (Fig.1(right)). The model resolution of the respective models does not play an important role in this case. While the HadGEM3 simulations have a strong positive bias, all the other

models tend to have a more negative bias. This is also reflected in the combined CREs as the biases in the LW tend to be relatively smaller. The model biases vary widely over the warm pool area in the western Pacific. While the HadGEM3 and MPIESM model simulations overestimate the TOA SW cloud radiative fluxes in the tropical monsoon belt, they underestimate the TOA LW fluxes by up to 15 W/m$^2$ . It is evident that both in DJF and JJA means, the opposite sign in the TOA SW and LW effects nearly compensates the fluxes over the tropics in the combined effects at the TOA.

As can be seen in Fig.1 when the zonal averaging of the CREs is performed, the differences between the high and standard resolution models remain low, mainly due to averaging out of over- and underestimations. Therefore, it looks as if the choice of the resolution does not seem to have a major impact in the simulation of the CREs. The regional differences emerge as we look in detail into the spatial patterns of the CRE differences between the Hi-res and Std-res set ups of the respective models. This is presented in Figs. 2 and 3 at the TOA (as an example) for the DJF and JJA months respectively. It can be

seen that the EC-Earth and the ECMWF models have lower differences, indicating insensitivity to the resolution. The Hi-res set-up of MPIESM model is, however, strongly overestimating the SW response by around 15 W/m$^2$ over the Southern Oceans compared to its corresponding standard resolution set up during DJF mean months. Though a slight underestimation by the Hi-res set up is observed in the HadGEM3 model over this region, the major difference is observed over the tropics, where the Hi-res set up is overestimating the SW CRE over the tropical Pacific and Indian oceans compared to the respective Std-res

set up. However, the Hi-res set ups of the respective models seem to underestimate the CREs globally in the LW compared to their standard resolution counterparts. The notable underestimation is over the equatorial west Pacific. While the Hi-res EC-Earth and ECMWF model set ups tend to slightly underestimate the LW CRE, the Hi-res HadGEM3 model set up tend to overestimate this over the South East Asian region. A completely different picture can be seen in the JJA mean CREs at the TOA (Fig. 3). Strong differences in the SW CREs are simulated in the MPIESM and HadGEM3 models, with significant

overestimation in Hi-res set ups over the North Pacific in the HadGEM3 models and north of 40N in the MPIESM model compared to their Std-res model counterparts. The impact of the resolution seems to be fairly negligible in the ECMWF model. The Hi-res setups of the respective models underestimate the LW CRE in general. This underestimation is prominent over South East Asia and equatorial Pacific in the EC-Earth and HadGEM3 models. Stronger response to increased resolution is simulated over southern India and northern Africa in the Hi-res HadGEM3 model.

It is noteworthy that the cloud regimes that seem to be affected by increasing resolution are different in different models. For example, in DJF, the HadGEM3 models show largest differences in the convective ITCZ regions, while MPIESM over the Southern Oceanic stratocumulus regions. The most drastic change in resolution is occurring in the HadGEM3 models (from 200 km to 50 km). This may have impact on SST resampling and thus convection. In the case of Southern Oceanic clouds, the increasing resolution in MPIESM may change the humidity PDFs (probability distributions functions) in a way that would

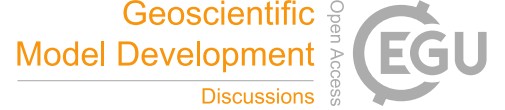



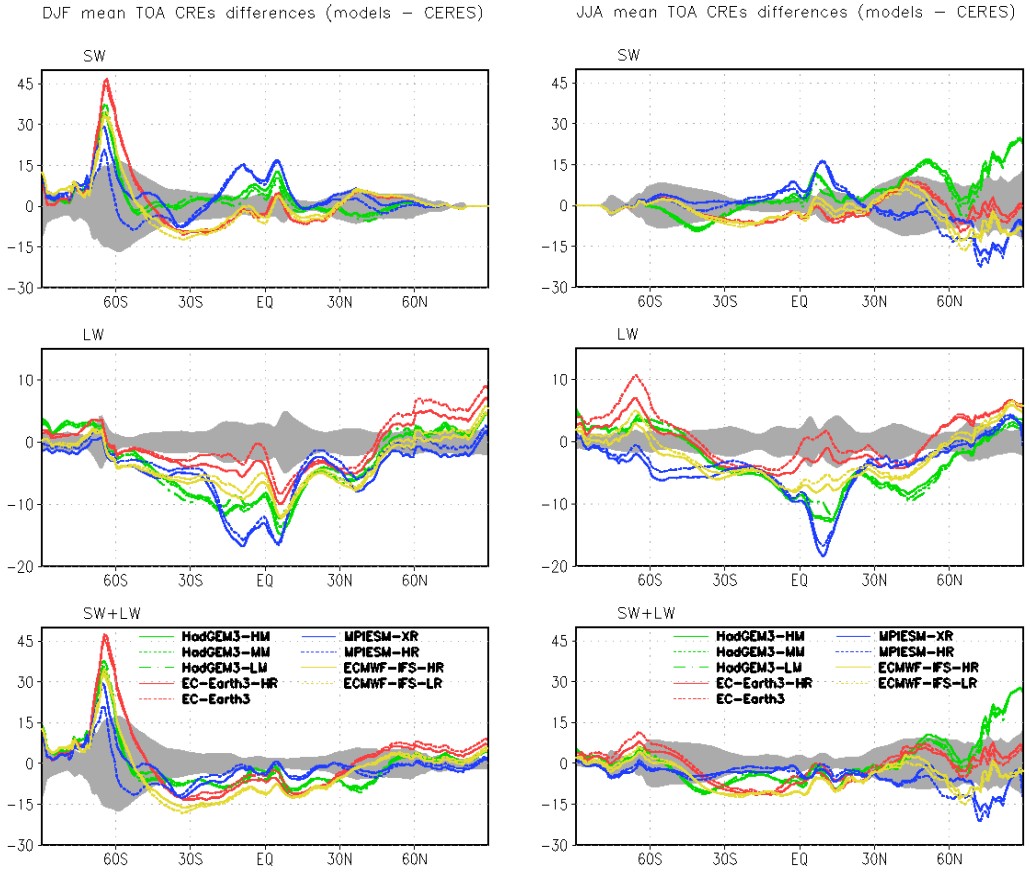

**Figure 1.** The model simulated SW, LW and combined TOA cloud radiative effects in W/m$^2$ shown as differences from the CERES-EBAF observations for (L) DJF mean and (R) JJA mean. The green lines correspond to the simulations with HadGEM3 model, red lines to EC-Earth3 model, blue lines to MPIESM model and ECMWF model are represented by yellow lines. The grey coloured envelope indicates one standard deviation of CREs based on CERES, shown here as a measure of natural variability in the observations.

change cloud fraction (since the relative humidity is already persistently high in this region). And the lack of tuning in higher resolution versions can further explain the observed differences.





**Figure 2.** Differences in the CREs at the TOA during DJF (W/m$^2$) between the High (Hi-res) and Standard resolutions (Std-res) of the respective models in SW (L) and LW (R).





**Figure 3.** Same as Fig.2, but, during JJA.



## 3.2 CREs at the surface

The differences in the model simulated CREs at the SFC from the observations are shown in Fig. 4 in SW, LW and SW+LW averaged over DJF (left) and JJA (right) months. A similar picture, as observed at the TOA, can be seen at the SFC in SW CREs over the Southern Ocean clouds in the DJF season. All the models show a positive bias over this region, similar in magnitude

as the TOA CREs. The MPIESM models tend to simulate a lower positive bias with the lower resolution set up reducing this bias even more. Over the tropical belt, the models exhibit a similar variability, but, a marginally stronger bias is simulated in SW CREs at the SFC compared to that what is seen at the TOA. A similar tendency as at the TOA is observed in JJA mean SFC CREs in tropics and beyond 30N. The differences are enhanced during DJF and JJA months in LW CREs at the SFC, when compared to that seen at the TOA. While all the models, irrespective of their resolutions, tend to simulate the LW CREs

reasonably well over the tropics in both seasons, large discrepancies can be seen at higher latitudes. A strong overestimation is simulated by all the models in LW CREs south of 60S and a strong underestimation north of 30N. The EC-Earth and ECMWF models simulate a lower positive bias in LW CREs compared to the other model set ups over the Southern Oceans in the mean DJF months. The JJA mean LW CREs are poorly simulated by all the models southward of 45S and northward of 60N. The biases in the SW CREs at the TOA and the surface are correlated, while less so in the LW. This is mainly due to the fact that

the LW CRE at the surface is heavily dependent on the cloud base heights and the surface conditions. Both of these factors do not change significantly in the models for optically thicker clouds. In comparison the different description of convection can heavily impact cloud top pressure and thus, the LW TOA CREs.

During polar summers in both hemispheres, strong biases are observed in the surface SW CREs. These biases are most pronounced over the regions where seasonal sea-ice melt drives the intraseasonal variability in sea-ice. The magnitude of these

biases can reach up to 40 W/m$^2$ over sea-ice regions near Antarctica and up to 30 W/m$^2$ over the Arctic Ocean. The signs of the biases are however different in the both hemispheres during their respective summers. While the models mostly tend to underestimate the SW CRE over the Arctic in NH summer, they tend to overestimate it over Antarctica in the SH summer. Having a correct description of surface albedo in models is crucial to minimize these biases. However, it is evident that the models differ considerably from observations and from one another as each model has its own formulation of sea-ice albedo

(Koenigk et al., 2014), for example, a climatological annual cycle is used in EC-Earth models. This in turn has an impact on the formation on clouds through air-sea interaction processes. The biases in the LW CREs are also high in the polar regions at the surface, most likely originating from the biases in describing dominant atmospheric processes such as the strength of temperature inversions and heat and moisture transport (Medeiros et al., 2011; Woods et al., 2017). The higher positive bias north of 60N in the HadGEM3 model simulations both in the SW CREs during JJA months at both the SFC and TOA results

in a much higher positive bias compared to the other models in the net CREs.



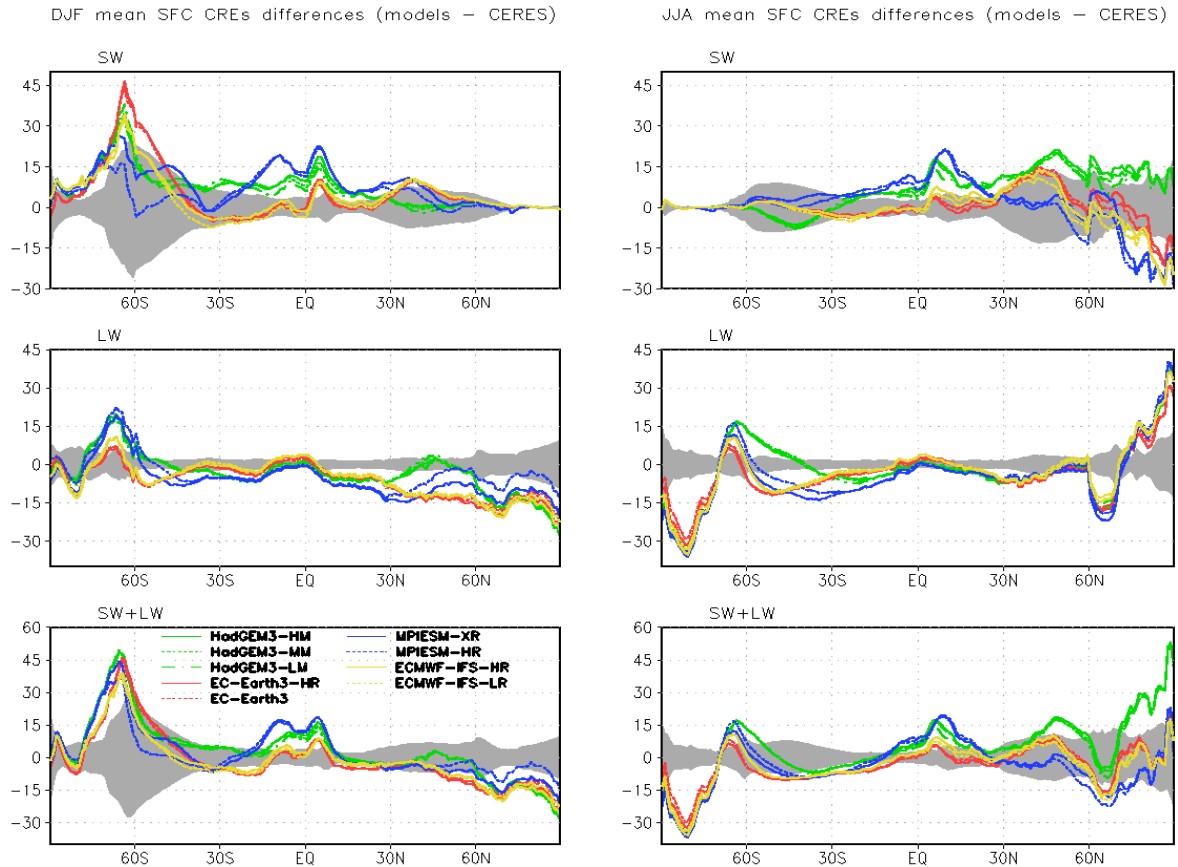

**Figure 4.** Same as Fig.1, but, at the surface.

## 4 Response of cloud radiative effects to ENSO

The TOA and SFC CREs associated with the ENP and ENN cases from model simulations are analyzed. The top row in Figure 5 first shows the CREs associated with ENP from CERES-EBAF observations at the TOA in SW (L) and LW (R). To investigate the model simulated responses, we split the models into high resolution model set ups and their standard resolution counterparts as explained in section 2.2. This would give us an understanding if increasing the spatial resolution results in an improvement of the response in the corresponding models. Hence, the second and third rows in Fig. 5 show the differences of the model ensemble mean of Hi-res and Std-res from the observations respectively and the intermodel differences are plotted in the bottom row. The intermodel differences are calculated as follows. At each grid point, if all models agree on the sign of the response, the absolute difference between the models showing the highest and lowest response is reported as the intermodel difference. Figure 6 shows the same, but, at the surface. Furthermore, Figures 7 and 8 show similar response, but during the ENN case at the TOA and SFC respectively.





## 4.1 The ENP case

In the ENP case, negative CRE anomalies (cooling) of up to 35 W/m$^2$ over western and central Pacific in the SW and positive anomalies (warming) of magnitude 20 W/m$^2$ over the same region in the LW at the top of the atmosphere are observed. This is expected, because, during the positive phase of the El Niño, the Walker circulation weakens resulting in warmer ocean surface

temperatures over the eastern and central Pacific which favors increased deep convective and stratiform clouds in this region and reduced cloud cover over the south east Asian regions (Fig. 9) and the opposite is observed during the La Nina phase (Eastman et al., 2011; Park and Leovy, 2004). This induces enhanced cooling/warming in SW/LW respectively, not only at the TOA, but also at the surface in the SW. The LW signal at the surface during ENP is considerably weaker, as for similar convective systems, the cloud base heights over the oceans do not change significantly in the models.

It is observed that the pattern correlations (i.e. the Pearson product-moment coefficient of linear correlation) with CERES observations in the tropical belt (30N-30S) are approximately above 0.75 for all the models irrespective of their resolution (not shown here). This suggests that the models realistically reproduce the spatial variability of the CRE response. However, the magnitude of this response and the location of the peak cooling/warming vary substantially regionally among the models as can be seen from the differences of the ensemble model means from the observations.

Both the model set-ups, i.e. Hi-res and Std-res, simulate the peak cooling region in SW cloud radiative fluxes at the TOA and surface over the western and central Pacific during ENP reasonably well. The multi-model ensemble mean strongly over-estimates the TOA and surface SW CREs north and south of the peak cooling region over the western Pacific by around 10 W/m$^2$ and underestimates the cooling by around 5 W/m$^2$ over central Pacific. The Hi-res model set ups simulate a stronger bias than the Std-res models over this region. The models, both the Hi-res and the Std-res models slightly overestimate the cooling

over the tropical Indian Ocean and underestimate the warming over SE Asia at the surface and at the TOA. Over the SE Asian region, the underestimation at the TOA is around 5-8 W/m$^2$, more so, in the Hi-res ensemble model mean.

The models, irrespective of their resolution, tend to simulate the peak ENSO response over central Pacific to the TOA LW cloud radiative fluxes reasonably well. The LW biases over south west Pacific are marginally stronger in the Hi-res models compared to the Std-res models. An opposite sign in the biases are observed in the LW CREs compared to the SW CREs at

the TOA. Although the model biases in LW at the TOA during the positive phase of ENSO are small, clear hemispherical differences can be seen at the TOA in the ENP case characterized by negative biases in the northern and positive biases in the southern hemisphere. Considering that the models do capture the broad spatial pattern in the CRE response, but, at the same time exhibit wave like structures in the SW biases and hemispheric nature of LW biases, it can be due to the fact that the shift of Walker circulation in the models is not followed with corresponding changes in cloud optical and physical characteristics.

The signal in the LW CREs at the surface during ENP is muted and hence, are the biases.

Large variability of up to 20 W/m$^2$ can be seen in the intermodel differences on the simulation of TOA and surface SW CREs associated with ENP and are mainly over the Pacific. The model bias is higher in the SW CREs compared to the LW CREs. The TOA biases are consistent with those observed in the set of CMIP5 models carried out with varying resolutions forced by AMIP SSTs (Wang and Su, 2015). These strong model over-/underestimation over the tropical convective regions




could be because of the discrepancies in the simulation of convective clouds (Wang and Su, 2013), as models have a tendency to produce optically thicker and deeper clouds compared to observations, whereas thin cirrus is prevalent in observations in those regions.

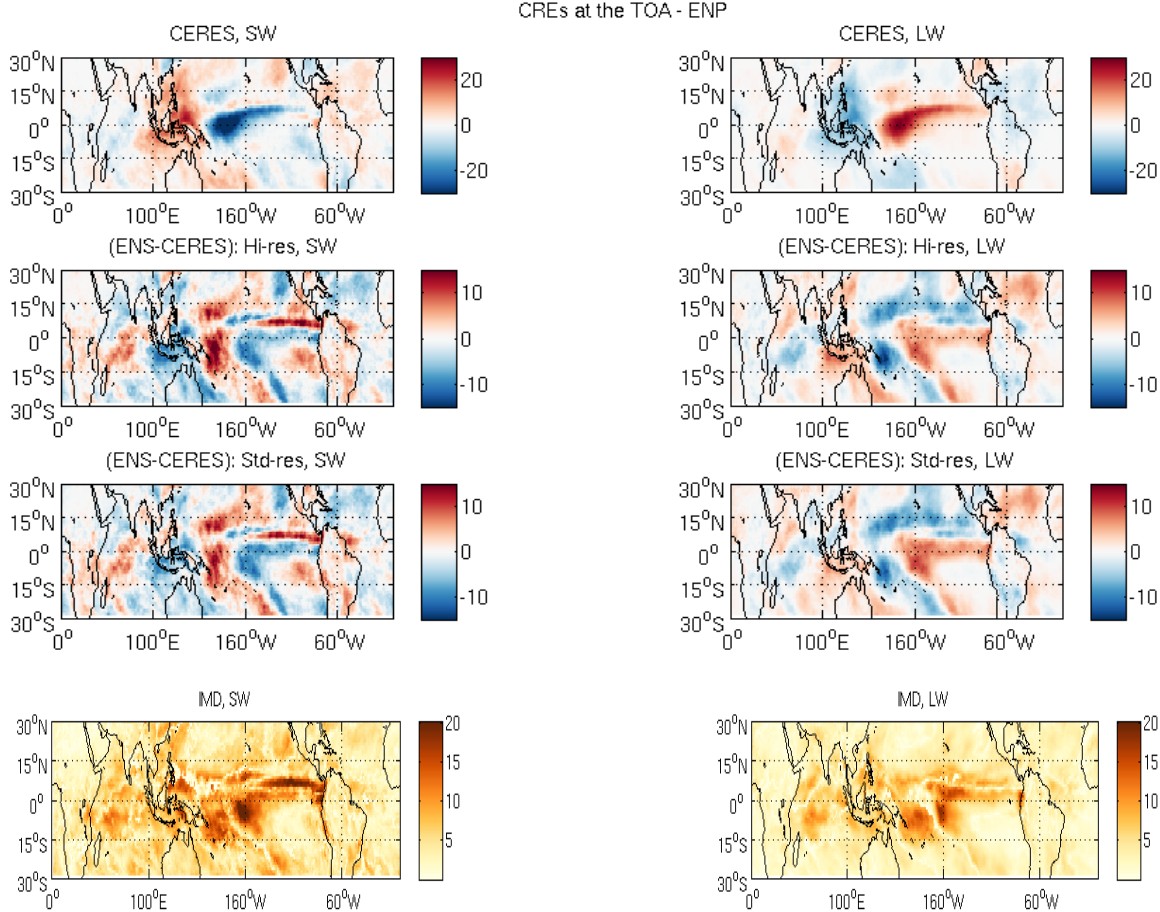

**Figure 5.** The SW (L) and LW (R) cloud radiative fluxes at the TOA as a response to positive phase of ENSO (ENP) from top row: CERES-EBAF observations, $2^{nd}$ row: the ensemble mean high resolution (Hi-res) and $3^{rd}$ row: standard resolution (Std-res) model simulated differences of this response from observations. Bottom row: the ensemble inter-model differences in W/m$^2$.





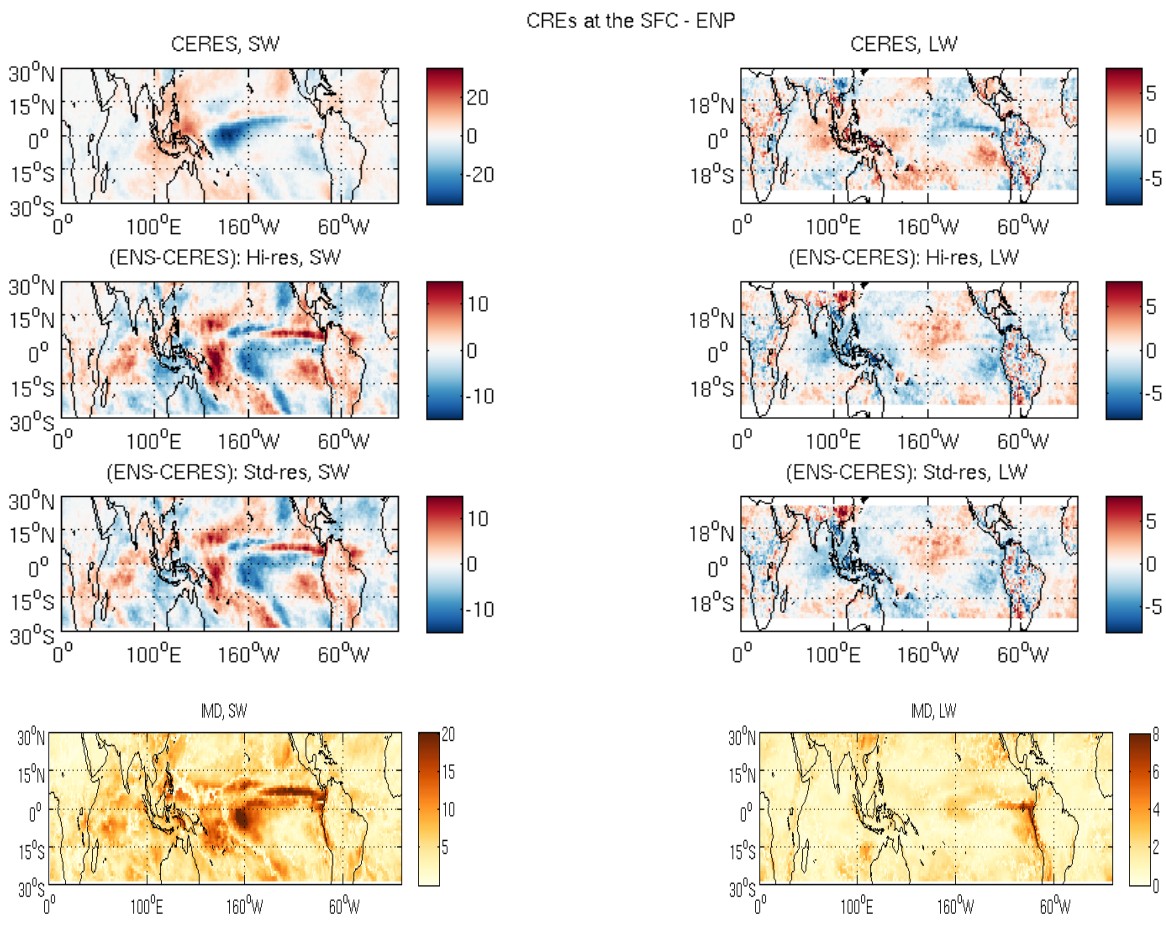

**Figure 6.** Same as above, but, for the cloud radiative response at the surface.

## 4.2 The ENN case

In the ENN case, a signal of opposite sign as that of ENP is observed with positive CREs in the SW at the TOA and the surface and negative anomalies in the LW at the TOA over the western and central Pacific. Over south east Asia, a weaker signal is observed. Though the models, marginally underestimate the warming in the SW associated with ENN at the TOA and the surface, they simulate the response in the LW at the TOA reasonably well. The Hi-res model set ups tend to slightly intensify this underestimation in the SW compared to the Std-res model set ups. The LW CRE associated with ENN at the surface is weaker. No notable improvements can be seen in simulating the cloud response in the LW using Hi-res model setups. The intermodel differences are smaller in the simulation of the TOA and surface LW CREs compared to the SW.



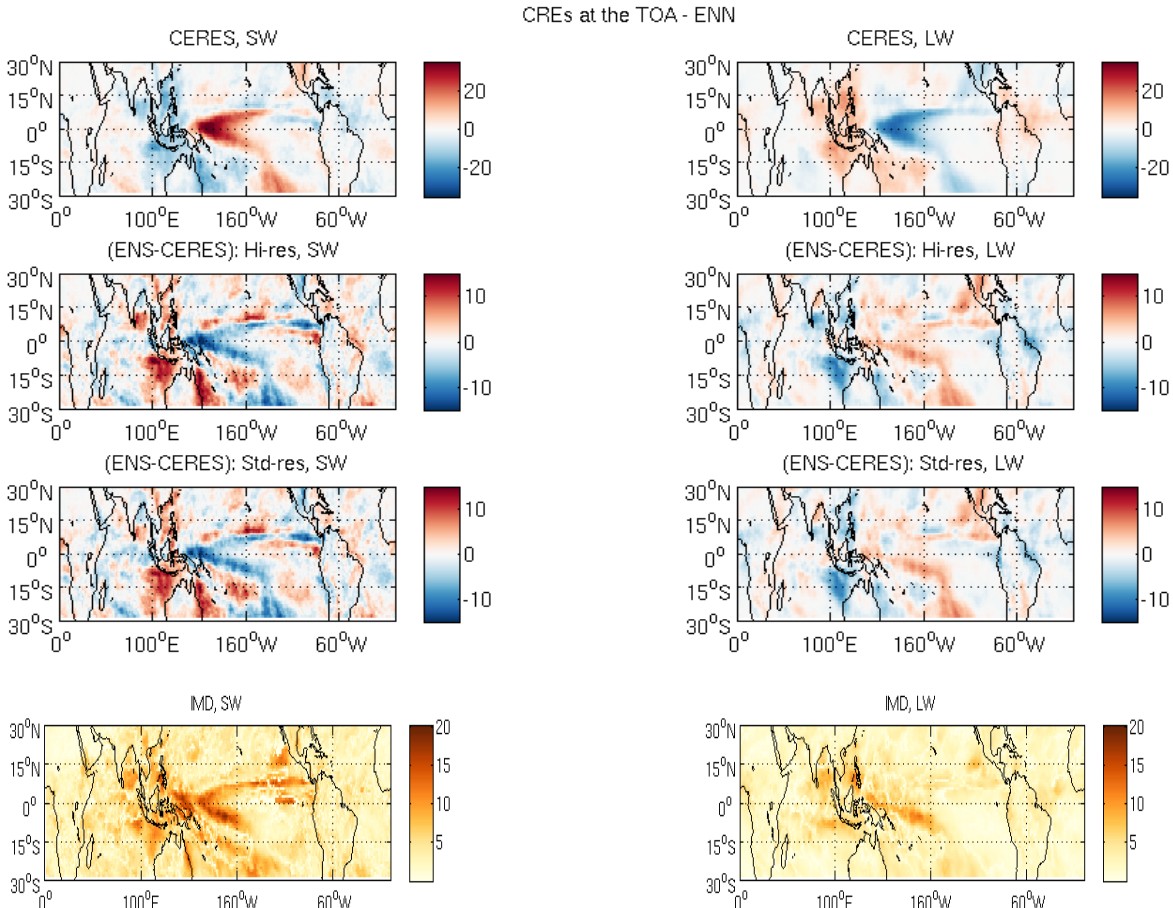

**Figure 7.** The SW (L) and LW (R) cloud radiative fluxes at the TOA as a response to negative phase of ENSO from top row: CERES-EBAF observations, $2^{nd}$ row: the ensemble mean high resolution (Hi-res) and $3^{rd}$ row: standard resolution (Std-res) model simulated differences of this response from observations. Bottom row: the ensemble inter-model differences in W/m$^2$.



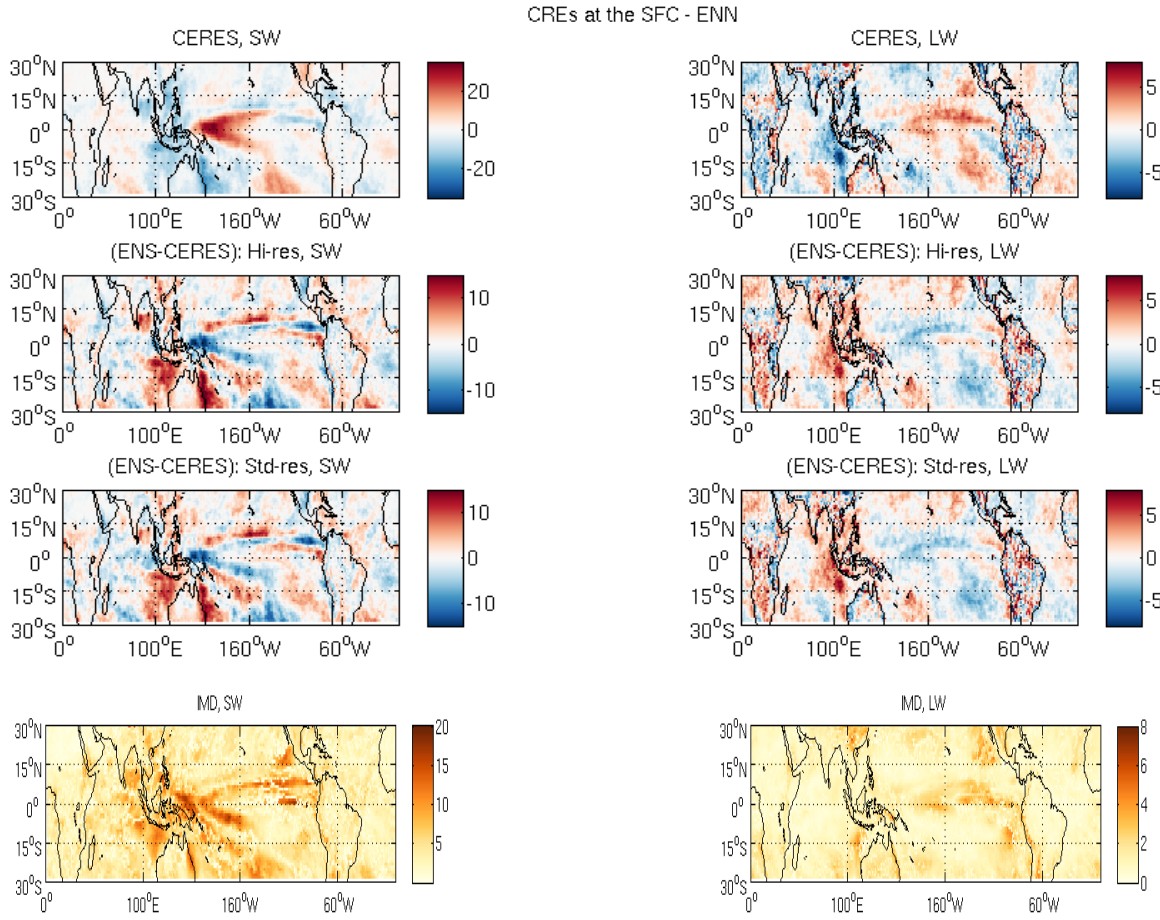

**Figure 8.** Same as above, but, for the cloud radiative response at the surface.

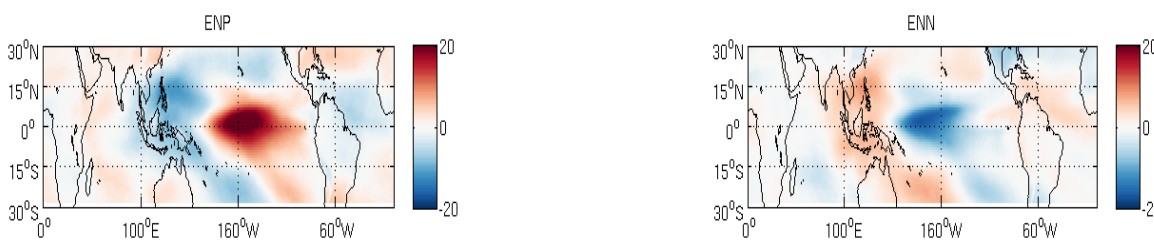

**Figure 9.** Ensemble model mean of simulated total cloud fraction anomalies (%) as a response to the positive phase of ENSO (L) and negative phase of ENSO (R).





## 4.3 The regional absolute biases

In order to better understand the role of varying spatial resolution locally, we further examine individual models under their high (HR) and standard (LR) resolution set ups. Fig. 10 shows the average absolute biases in TOA and SFC CREs in SW and LW during the positive and negative phases of ENSO, with reference to CERES, over Niño3.4 region (170W-120W, 5N-5S).

5  It can be seen that the absolute biases across the models are high in SW at the TOA and SFC and in LW at the TOA during the positive phase of ENSO, particularly in the HadGEM and EC-Earth models. The uncertainty bars show one standard deviation in the CERES anomalies for the respective cases as a measure of the variability in the observation data. It is to be noted that, in all cases, the observed biases over the selected region remain below the variability in the CERES data. The HR set ups of HadGEM3 and EC-Earth models have a lower bias compared to their LR set ups. The opposite is seen in MPIESM models.

10  ECMWF model, irrespective of their resolution, show a similar bias.

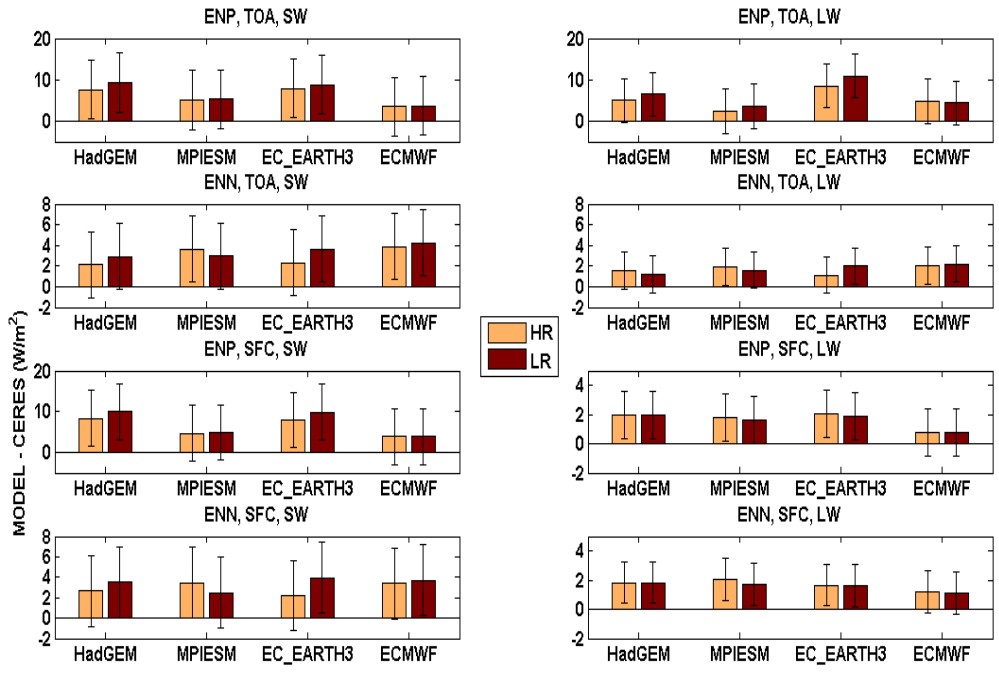

**Figure 10.** Average absolute errors in the high (HR) and standard resolution (LR) resolution of the model versions with reference to CERES averaged over the Niño3.4 region (170W-120W, 5N-5S) in SW (L) and LW (R) during the enhanced positive and negative phases of El Niño at the TOA (rows 1-2) and at the SFC (rows 3-4). The uncertainty bars show one standard deviation in the CERES anomalies for the respective cases as a measure of the variability in the observation data.





## 5  Response of cloud radiative effects to NAO

The TOA and SFC CREs associated with NAOP from observations are shown in top row of Figs. 11 and 12 respectively. The second and third rows show the ensemble mean difference of the response simulated in Hi-res and Std-res models with the observations respectively and the intermodel differences are shown in the bottom row. Figs. 13 and 14 show the same, but for the NAON case. The SW (left) and LW (right) components of the total CREs are shown separately in each of the NAOP and NAON cases.

### 5.1  The NAOP case

During the positive phase, as the polar vortex strengthens trapping the cold air in the central Arctic, the winter storms in the North Atlantic penetrate further to the north, with its remnants reaching deep over the northern Norwegian and Greenland Seas. The North East Atlantic is usually, persistently cloud covered. However, the additional transport of heat and moisture brought about by winter storms leads to increased opacity of these cloud systems. This is evident in the slight decrease in the TOA and surface SW CRE. The increased opacity of clouds leads to additional reflection of solar radiation back to the space, while the clouds also emit at warmer temperatures than normal. LW CRE is especially stronger over the Scandinavia and Norwegian Sea at the TOA. The LW CRE anomalies over the North Atlantic are quite muted at the surface, whereas the Greenland and Canadian sectors of the Arctic show increased LW CREs. This is mainly due to the fact that, in contrast to open oceanic waters in the North Atlantic, clouds can exert strong LW forcing over the ice and snow covered areas in the Arctic. Over the Mediterranean region and Iberian peninsula, colder and drier conditions prevail during the NAOP case due to the northward shift of the storm tracks. This results in a significant reduction in cloud cover over this region, as can be seen in the model simulated NAO related total cloud fraction anomalies during this phase (Fig. 15) which is also in consistency with the previous studies (Chaboureau and Claud, 2006; Trigo et al., 2002). Clearer conditions result in an increase in SW CRE and a decrease in LW CRE at the TOA and at the surface.

The Hi-res and Std-res model ensemble mean differences against CERES observations are generally quite low (below $\pm 5$ W/m$^2$) and do not exceed one standard deviation of the CRE anomalies observed in the CERES data over the majority of the regions (not shown). The models capture the spatial cloud radiative response to the positive phases of the NAO quite well. For example, the models, irrespective of their resolution, simulate the response reasonably well over the North Atlantic, over Scandinavia and over the Meditteranean at the TOA in both SW and LW and at the SFC in SW. The models overestimate the cooling by 3-4 W/m$^2$ over continental Europe in the SW at the TOA and SFC. The LW TOA CRE is, on the other hand, underestimated over this region. However, strong discrepancies can be noted in the SFC LW CREs with models overestimating the response by more than 5 W/m$^2$ over northern Europe. Strong underestimation of similar magnitude in the LW CRE at the surface can be noted in the Canadian sector of the Arctic Ocean, and also over Greenland.

The CRE biases in the Hi-res and Std-res model set ups do not seem to be strikingly different from one another at a first glance. However, a significant improvement in the SW bias over eastern Europe in the Std-res simulations at the SFC during NAOP is seen compared to that at the TOA, whereas, the Hi-res simulations better simulate the TOA LW CREs over continental



Europe. No notable improvement is seen in the SFC LW CREs with resolution. The inter-model differences are of the same magnitude as that of the under- and overestimations in the CRE reponse at the TOA. The SW and LW biases are respectively much higher at the surface over continental Europe and over Scandinavia.

**Figure 11.** The SW (L) and LW (R) cloud radiative fluxes at the TOA as a response to positive phase of NAO from top row: CERES observations, $2^{nd}$ row: the ensemble mean high resolution (Hi-res) and $3^{rd}$ row: standard resolution (Std-res) model simulated differences of this response from observations. Bottom row: the ensemble inter-model differences in W/m$^2$.



**Figure 12.** Same as above, but, for the cloud radiative response at the surface.





## 5.2   The NAON case

In the NAON case (Fig. 13), the winter storms are not as intense and do not penetrate deeper into the northern North Atlantic as the cold air outbreaks from the Arctic over the northern high and mid-latitudes prevail, shifting the zonal temperature gradient southwards. As a result, the TOA SW CRE is higher than the usual over northern midlatitudes and the TOA LW CRE is lower than usual as the clouds emit at the colder temperatures, especially over Scandinavia. The LW CRE at the surface is decreased over Greenland and the Canadian Arctic and increased over the Eurasian Arctic (Fig. 14). This response is opposite to that observed in the NAOP case. The CRE response in the Mediterranean region is also, as expected, opposite to that of the NAOP case.

Both at the TOA and the SFC, though the biases are small in the SW CREs over the Atlantic and Scandinavia, the models underestimate the SW CREs over continental Europe by -4 W/m$^2$. The models simulate the LW CREs at the TOA reasonably well, however, marginal underestimation in the cooling in the North Atlantic in LW CREs at the TOA can be noted. At the SFC, the biases in the LW CREs are highest over Northern continental Europe, Greenland and along west coast of Norway, but are of opposite sign, in that the models underestimate CREs over northern Europe and west coast of Europe and overestimate it over Greenland (locally exceeding 5 W/m$^2$). Over the Eurasian and Canadian Arctic regions, the biases in the surface LW CREs are of opposite sign that of the NAOP case.

An improvement in the SW CREs at the TOA can be noted in the Hi-res model ensemble mean over continental Europe at the TOA and SFC. Though there is a marginal improvement in the LW CREs at the TOA over North Atlantic, no notable differences are seen at the SFC. The inter-model differences, like in the case of NAOP, are much higher in the SW than the LW at the TOA and SFC, particularly over continental Europe. The differences are the same or even less in magnitude as that of the under- and overestimations of the CRE response.





**Figure 13.** The SW (L) and LW (R) cloud radiative fluxes at the TOA as a response to negative phase of NAO from top row: CERES observations, $2^{nd}$ row: the ensemble mean high resolution (Hi-res) and $3^{rd}$ row: standard resolution (Std-res) model simulated differences of this response from observations. Bottom row: the ensemble inter-model differences in W/m$^2$.







**Figure 14.** Same as above, but, for cloud radiative response at the surface.



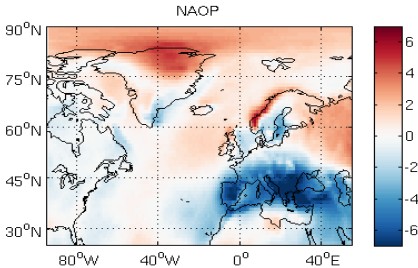 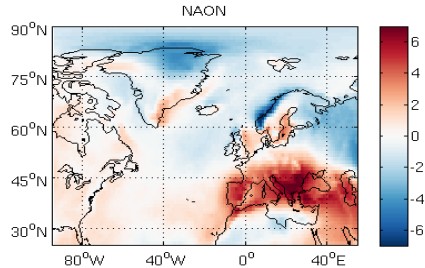

**Figure 15.** Ensemble mean of model simulated total cloud fraction anomalies (%) as a response to the (L) positive phase of NAO and (R) negative phase of NAO.

## 5.3 The absolute regional biases

Fig. 16 shows the absolute biases in the high (HR) and standard resolution (LR) model set-ups for the different phases of NAO over Europe (40W-40E, 30N-75N). This region is active with winter storms during the positive phases of the NAO, which eventually transport heat and moisture to the northernmost latitudes. Here, it has to be noted that the biases are comparatively smaller than the biases observed in the ENSO cases. Further, there is no noticeable improvement with increased resolution. This indicates that improving the surface description and treatment (e.g. surface snow and ice variability) in models might be more important than increasing only the horizontal resolution for cloud processes. The uncertainty bars show that the biases over the selected region remain below the variability in the CERES data. This means that the model biases are not significant.





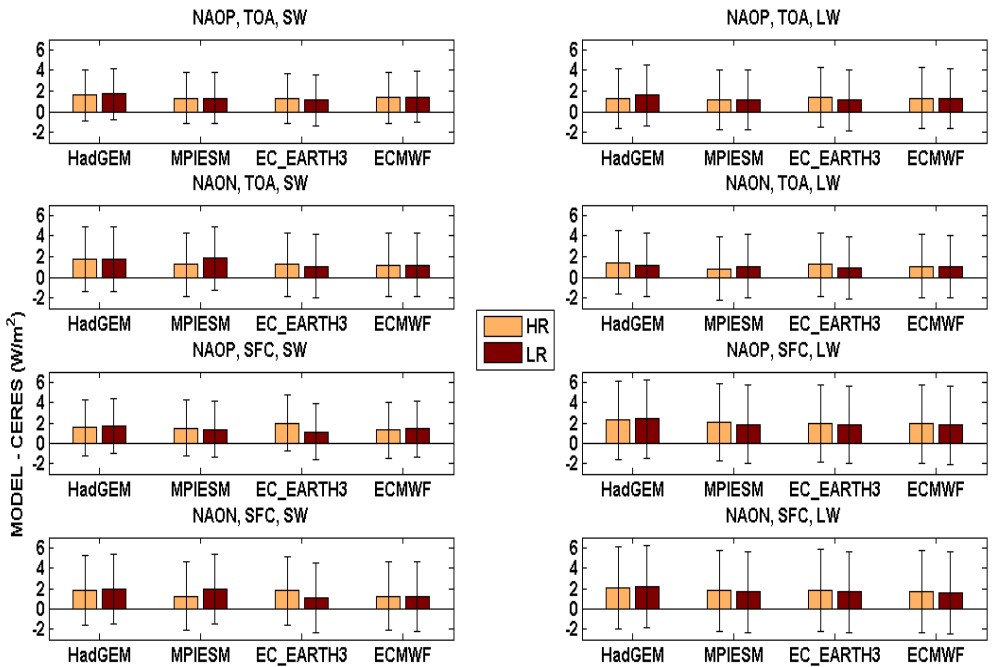

**Figure 16.** Average absolute errors in the high (HR) and low resolution (LR) of the model versions with reference to CERES averaged over Europe (40W-40E, 30N-75N) in SW (L) and LW (R) during the enhanced positive (NAOP) and negative (NAON) phases of NAO at the TOA (rows 1-2) and at the SFC (rows 3-4). The uncertainty bars show one standard deviation in the CERES anomalies for the respective cases as a measure of the variability in the observation data.



## 6   Conclusions

In the present study, we evaluated 4 global climate models at different spatial resolutions to assess how well they simulate CREs, both at the top of the atmosphere and at the surface, as well as their shortwave and longwave components. The focus is placed on evaluating cloud radiative response to two leading modes of natural variabilities, namely ENSO and NAO, allowing

process-oriented evaluations. The simulations from the high and low resolution model set-ups were contrasted to investigate if any value can be added by increasing the spatial resolution of the different models. The retrievals of CREs from CERES instruments onboard a series of satellites were used as the observational reference. The following conclusions can be drawn from the evaluations.

a) The largest disagreements between models and observations occur over the polar regions, both at the TOA and the SFC,

and especially over the locations where seasonal sea-ice variability is strongest. The surface SW CRE plays an important role during the melt season. The models, however overestimate this forcing by up to 35 W/m$^2$ over coastal Antarctic and underestimate it by 20-30 W/m$^2$ over Arctic. This will have an implication for quantifying the cloud feedbacks on the sea-ice and estimating future changes in sea-ice during the melt season.

b) The zonally averaged CREs do not seem to be resolution dependent. This means that all the models follow a similar

response irrespective of the resolution at most regions. However, regional differences emerge when looking at the spatial patterns of the forcings. Here, it is seen that different cloud regimes are affected by increasing resolution in different models.

c) The spatial patterns of cloud radiative response to ENSO in the tropical belt is simulated reasonably well by the models, with spatial correlations up to 0.75. However, strong biases in the magnitude of this response is noted. The model biases are generally half as that of the actual cloud radiative response seen in the CERES data for the ENSO cases (5-10 W/m$^2$) at both

TOA and the surface, with Hi-res model set ups simulating a stronger bias than the respective Std-res models. The biases in the LW CRE tend to be smaller than in the SW CRE. The intermodel differences in the SW CRE at the TOA and surface over the convectively active regions are stronger, nearly of the same order as the actual response. The intermodel differences in the LW CRE are lower at the surface during both ENP and ENN, typically within a few W/m$^2$. This suggests that the parameterization of SW radiative transfer and the treatment of cloud optical properties vary strongly among the models. The

large-scale organization of convection and associated cloud types can also be different.

d) In the case of NAO, the model biases are less than observational uncertainties and also well within the observational variability (less than one-sigma) in the CREs. The spatial patterns of the response are also simulated quite well by the models during the positive and negative phases of the NAO. The biases in the surface LW CREs have a strong meridional character, in that the they are of opposite sign over the eastern and western parts of the Arctic across 20W, and also have opposite sign to

that of the cloud radiative response observed in the CERES data.

e) The average absolute biases over the Niño3.4 region for the ENP and ENN cases and over Europe (40W-40E, 30N-75N) for the NAOP and NAON cases are investigated in the high and low resolution of each model. The absolute biases in both the cases are well below the uncertainty in the observational data. The average biases in the case of NAO are smaller than the biases seen over the Niño3.4 region. The Hi-res set up of HadGEM and EC-Earth models has a lower bias compared to their





Std-res counterparts over the Niño3.4 region, whereas, an opposite signal is seen in MPIESM models. ECMWF model set ups exhibit the same biases irrespective of the resolution.

*Data availability.* Access to the model output data used in this study will be available through the European Research Council Horizon 2020 PRIMAVERA project (https://www.primavera-h2020.eu/modelling/data-access/). More information regarding model configurations and data
5 availability are available from the authors upon request.

*Acknowledgements.* This study was financially supported by PRIMAVERA (PRocess-based climate sIMulation: AdVances in high resolution modelling and European climate Risk Assessment), a Horizon 2020 project funded by the European Commission.



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
