# Peer review of "A statistical and process oriented evaluation of cloud radiative effects in high resolution global models"

_Geoscientific Model Development, 2018_

## Referee Comment (RC1) · Anonymous Referee #1 · 9 Jan 2019

**General comments**

The paper investigated the cloud radiative effects in simulations with different spatial resolution for 4 climate models, which are part of the HighResMIP, compared to CERES observation. The results are presented for different seasons and modes of internal climate variability (ENSO and NAO) and show that an increased spatial resolution does not explain most differences to the observations. The paper is well written, comprehensible, and suitable for publication in GMD, but I have some minor issues (and technical corrections) that are listed below.

Specific comments

Page 3, line 10/ page 4 line 31 - page 5, line 2 The division in high and low/standard resolution models should be explained already in section 2.1 (page 3 about line 10) instead of at the end of section 2.3 because it is use throughout the complete paper, not only for the ENSO analysis. It would also be better to introduce the abbreviations (Hi-res, Std-res or HR, LR) there and then only use one kind, not e.g. Hi-res (mainly used), high resolution (page 11, line 4), and HR (used in section 4.3 and 5.3).

Page 6, line 11 I think it is too early to draw the conclusion here that differences for zonally averaged hi-res and std-res models remain low, "mainly due to averaging out of over- and underestimation". At this point this should be rather written as assumption or question. Especially since it seems not to be true for all models according to page 6, line 26: "The impact of the resolution seems to be fairly negligible in the ECMWF model."?

Page 10 For the CREs at TOA there is a detailed discussion of the spatial (meridional) pattern (shown in Fig. 2 and 3), but these pattern are not at all mentioned for the CREs at the surface in Section 3.2. While it is probably not necessary to show similar figures like Fig. 2 and 3 also for the surface (such figures could be include in a supplement, though) it should be mentioned if the spatial pattern are in general similar to the ones at TOA or if there are any striking differences?

Page 11, line 8ff The calculation of the intermodel differences should be explained in more detail. What happens if not all models agree on the sign? Especially in Fig 13 and 14 there are some surprising low values in areas with high values (off the US East-coast), are these missing values/ values where the sign does not agree? If yes they should be marked by an easier to distinguish color.

Page 26, line 8 Here, it should be also mentioned, that the study is based on atmosphere only simulations as stated in the last sentence of the Abstract: "However, we note that these results are obtained from atmosphere-only simulations and the impact of changes in atmospheric resolution may be different in the presence of coupled climate feedbacks."

**Technical corrections**

There are three different ways to write "setups" ("set ups", "set-ups") in the paper?

Page 2, line 22-25: The sentence is hard to understand, maybe there is something missing? Better split it in two sentences.

Page 3, line 8: Maybe better "... simulations. The forcing includes ..." instead of "... simulations that include ..." would be easier to understand?

Page 3, line 16: "The CREs ... are defined ..." instead of "The CREs ... is defined ..."

Table 1 and several places in the text: Shouldn't it be "MPI-ESM" instead of "MPIESM"?

Page 4, line 20: "... associated solely with ..." instead of "... associated solely by ..."?

Page 5, line 14-15: This explanation would not be necessary if the separation between Hi-res and Std-res would be introduced in section 2.1 as mentioned in the specific comments.

Page 6, line 21: "most notable" instead of "notable"

Page 6, line 34: "probability density function" is the more common?

Fig. 1 and 4: Add a y-label?

Fig. 2 and 3: Use only one color bar and try to enlarge the size of the panels?

Page 11, line 4-5: At the moment it is explained in section 2.3 (not 2.2) and I think it should be explained in section 2.1, see specific comments.

Page 12, line 34: The abbreviation "AMIP" is not explained.

Fig. 5 and 7, caption: There is no reason to explain the abbreviations Hi-res and Std-res again?

СЗ

Page 26, line 28-30: There is something wrong in this sentence, probably an extra "the" at "that the they are"?

Page 26, line 33: Following the discussion in section 5.3 it should be rather "variability" than "uncertainty"?

---

## Referee Comment (RC2) · Anonymous Referee #2 · 11 Mar 2019

Overview:

The goal of this model evaluation study is to compare the cloud radiative effects between standard-resolution and high-resolution climate models. The authors find that cloud radiative effects differ strongly between models, but not nearly as much between runs with different resolutions of the same model. The authors conclude that the apparent insensitivity to increased atmospheric horizontal resolution indicates that physical parameterizations play a dominant role in determining the behavior of cloud-radiation feedbacks.

The manuscript touches an important topic and is worthy of publication. I did not find
any major flaws that would require substantive revisions. The article could benefit from a final discussion on what the findings mean for model development (e.g., Should we focus on better parameterizations? Is it useful to run high-resolution climate models if biases aren't really improved compared to lower resolution models?)

General Comments:

1. Especially the abstract and the beginning of the manuscript are well written. However, the manuscript becomes tedious to work through after about page 6. I think this may be just a reflection of the topic, since model validation studies tend to be tedious.

2. One of the points that makes the article tedious to read is that the authors switch between "low resolution" and "std-res" models. It would be easier if just one qualifier is chosen, in this case probably std-res (standard resolution).

3. The quality of the figures could be better. For example, the multi-panel map plots have lots of white space between the two columns, and lots of space is occupied by large colorbars. The individual panels could be enlarged at the expense of the wasted space.

Specific comments:

- 1. page 1, line 10: no comma after whereas
- 2. page 1, line 18: EU-funded
- 3. page 4, line 23: add "the" before Niño3.4 index
- 4. page 26, lines 11-12: add "the" before coastal Antarctic and Arctic

GMDD

---

## Author Comment (AC1) · 22 Mar 2019

**Response to Reviewer #1**

Thank you very much for your remarks. We have tried to incorporate all your suggestions. Please find below a point by point response to your comments.

**Specific comments:**

Page 3, line 10/ page 4 line 31 - page 5, line 2 The division in high and low/standard resolution models should be explained already in section 2.1 (page 3 about line 10) instead of at the end of section 2.3 because it is use throughout the complete paper, not only for the ENSO analysis. It would also be better to introduce the abbreviations (Hi-res, Std-res or HR, LR) there and then only use one kind, not e.g. Hi-res (mainly used), high resolution (page 11, line 4), and HR (used in section 4.3 and 5.3).

The following sentence has been added to Section2.1 and has been deleted from section2.3 and section 2.4.
"For the analysis, the models are separated into high resolution (Hi-res) and standard resolution (Std-res) model configurations. The models that are included in the Hi-res are HadGEM3-GC31-HM, EC-Earth3-HR, MPI-ESM-XR and ECMWF-HR. Their respective low/standard resolution counterparts constitute the Std-res". We have standardized the abbreviations, Hi-res and Std-res throughout the manuscript.

Page 6, line 11 I think it is too early to draw the conclusion here that differences for zonally averaged hi-res and std-res models remain low, "mainly due to averaging out of over- and underestimation". At this point this should be rather written as assumption or question. Especially since it seems not to be true for all models according to page 6, line 26: "The impact of the resolution seems to be fairly negligible in the ECMWF model."?

This statement was made to convey that when one only look into the zonal averages, where we have plotted the CREs at the TOA from both Hi-res (solid lines) and Std-res (dashed lines) of the different models used in this study, the differences between these resolutions remain low. However, the spatial patterns reveal these differences more vividly. Hence, the spatial plots were added.

Page 10 For the CREs at TOA there is a detailed discussion of the spatial (meridional) pattern (shown in Fig. 2 and 3), but these pattern are not at all mentioned for the CREs at the surface in Section 3.2. While it is probably not necessary to show similar figures like Fig. 2 and 3 also for the surface (such figures could be include in a supplement, though) it should be mentioned if the spatial pattern are in general similar to the ones at TOA or if there are any striking differences?

Following the reviewer suggestion, the spatial patterns (for DJF and JJA means) of the differences in CREs at the surface are now plotted and added to the supplement. The following text is added to the manuscript mentioning the similarities and differences. "Similar to the TOA, the differences in spatial distribution in the SW CREs between the Hi-res and the Std-res model configurations are analyzed at the surface and are shown in Fig.A1 and Fig.A2 in Appendix-A for mean DJF and JJA respectively. It can be seen that the differences at the surface are similar, both spatially and in magnitude to what is seen at the TOA in winter. However, large differences are seen in the surface LW CREs. As in the case of the TOA, the ECMWF model is insensitive to a change in resolution. The Hi-res set up of the MPI-ESM model significantly underestimates the LW CREs north of 40N compared to its Std-res configuration. The DJF mean LW CRE biases are much smaller in EC-Earth3 model, but, the Hi-res set up overestimates the LW forcing over the oceans and underestimates over the continents. A strong overestimation is also seen in the Hi-res set up of

HadGEM3 model over the Southern Oceans and Eurasia. In summer, the SW CREs at the surface follow the same pattern as is seen at the TOA. However, the summer LW CRE biases at the surface are considerably weaker as compared to in winter. "

Page 11, line 8ff The calculation of the intermodel differences should be explained in more detail. What happens if not all models agree on the sign? Especially in Fig 13 and 14 there are some surprising low values in areas with high values (off the US East-coast), are these missing values/ values where the sign does not agree? If yes they should be marked by an easier to distinguish color.

We would really like to thank the reviewer for raising this issue. This is because, while revising the relevant figures showing intermodel differences (IMD), we discovered an inconsistency in our assumption, wherein the IMDs were calculated in the majority of the model set ups (i.e. 5 or more) agreeing on the sign of the bias instead of all the 9 model set ups agreeing. This is now corrected. The areas where all 9 model set ups do not agree in sign are marked with grey colour in the revised figures.

The revised text now reads as:
"The intermodel differences are calculated as follows. At each grid point, if all 9 model set ups agree on the sign of bias with respect to the CERES observations, the absolute difference between the model set ups showing the highest and lowest bias is reported as the intermodel difference. The regions, where all 9 model set ups do not agree in the sign of the bias, are marked in grey colour."

Page 26, line 8 Here, it should be also mentioned, that the study is based on atmosphere only simulations as stated in the last sentence of the Abstract: "However, we note that these results are obtained from atmosphere-only simulations and the impact of changes in atmospheric resolution may be different in the presence of coupled climate feedbacks."

It is clarified in the revised manuscript. A final discussion is added to the 'Conclusions' section to sum up the main results.

**Technical corrections**
There are three different ways to write "setups" ("set ups", "set-ups") in the paper?

"set ups" is now consistently used throughout the manuscript.

Page 2, line 22-25: The sentence is hard to understand, maybe there is something missing? Better split it in two sentences.

The sentence is rephrased in the revised manuscript as,
"Here, in the context of this PRIMAVERA project, the surface and top of the atmosphere cloud radiative effects (CREs) are analyzed in global climate models from four European modelling centers, each with varying spatial resolutions. The observed flux estimates from NASA's CERES-EBAF (Clouds and the Earth's Radiant Energy System-Energy Balanced And Filled) instrument are used for the evaluation."

Page 3, line 8: Maybe better "… simulations. The forcing includes …" instead of "… simulations that include …" would be easier to understand?

The sentence is rephrased as,

"The atmosphere-only simulations are forced by SST and sea ice concentrations from the HadISST2.2 \citep{ken17} dataset."

Page 3, line 16: "The CREs ... are defined ..." instead of "The CREs ... is defined ..."
Table 1 and several places in the text: Shouldn't it be "MPI-ESM" instead of "MPIESM"?
Page 4, line 20: "... associated solely with ..." instead of "... associated solely by ..."?

All these details are corrected in the revised manuscript.

Page 5, line 14-15: This explanation would not be necessary if the separation between Hi-res and Std-res would be introduced in section 2.1 as mentioned in the specific comments.

The following sentence has been added to Section2.1 and has been deleted from section2.3 and section 2.4.
"For the analysis, the models are separated into high resolution (Hi-res) and standard resolution (Std-res) model configurations. The models that are included in the Hi-res are HadGEM3-GC31-HM, EC-Earth3-HR, MPI-ESM-XR and ECMWF-HR. Their respective low resolution counterparts constitute the Std-res. "

Page 6, line 21: "most notable" instead of "notable"
Page 6, line 34: "probability density function" is the more common?

The above suggestions are incorporated in the revised manuscript.

Fig. 1 and 4: Add a y-label?
Fig. 2 and 3: Use only one color bar and try to enlarge the size of the panels?

The figures are revised to incorporate the above-mentioned suggestions.

Page 11, line 4-5: At the moment it is explained in section 2.3 (not 2.2) and I think it should be explained in section 2.1, see specific comments.

Rephrased as,
"To investigate the simulated responses, the ensemble mean of the Hi-res and Std-res model configurations is analyzed."

Page 12, line 34: The abbreviation "AMIP" is not explained.
Fig. 5 and 7, caption: There is no reason to explain the abbreviations Hi-res and Std-res again?

This is modified in the revised manuscript.

Page 26, line 28-30: There is something wrong in this sentence, probably an extra "the" at "that the they are"?
Page 26, line 33: Following the discussion in section 5.3 it should be rather "variability" than "uncertainty"?

This is modified in the revised manuscript.

---

## Author Comment (AC2) · 22 Mar 2019

**Response to anonymous Referee #2**

Thank you very much for your remarks. We have tried to incorporate all your suggestions. Please find below a point by point response to your comments.

**Overview:**
The goal of this model evaluation study is to compare the cloud radiative effects between standard-resolution and high-resolution climate models. The authors find that cloud radiative effects differ strongly between models, but not nearly as much between runs with different resolutions of the same model. The authors conclude that the apparent insensitivity to increased atmospheric horizontal resolution indicates that physical parameterizations play a dominant role in determining the behavior of cloud-radiation feedbacks. The manuscript touches an important topic and is worthy of publication. I did not find any major flaws that would require substantive revisions. The article could benefit from a final discussion on what the findings mean for model development (e.g., Should we focus on better parameterizations? Is it useful to run high-resolution climate models if biases aren't really improved compared to lower resolution models?)

A following paragraph is added to the end of the 'Conclusions' section:
"From this study it is clear that the well-known issue of the large biases in SW CREs over the polar regions during the melt season does not improve by increasing the resolution of the models chosen here. This would require improvements not only in the parameterization schemes involving the microphysical properties of clouds, but also in the surface description. Analysis of the spatial pattern of the TOA SW CREs during winter reveal that different cloud regimes are affected drastically with a change in resolution in MPI-ESM and HadGEM3 models. For example, the Hi-res HadGEM3 model show an overestimation over the convective ITCZ regions compared to its Std-res counterpart and this may have an impact on SST resampling and thus convection. On the other hand, the Hi-res MPI-ESM overestimates the CREs over the Southern Oceanic stratocumulus region and this may have an impact on the cloud fraction. The observed differences can be attributed to the lack of tuning in higher resolution versions. Though the models tend to simulate the spatial variability in cloud radiative response to ENSO and NAO variability, they vary widely in the magnitude of the response. The CRE biases associated with the NAO phase are smaller compared to those with the ENSO phase. Although some improvements can be seen regionally, it is difficult to identify patters that hold across all models. Hence, it can be concluded that improving the physical parameterization schemes rather than increasing the resolution is perhaps important in better simulating the CREs. However, it has to be noted that these are atmospheric only simulations and the impact may be different in the presence of coupled climate models."

**General Comments:**
1. Especially the abstract and the beginning of the manuscript are well written. However, the manuscript becomes tedious to work through after about page 6. I think this may be just a reflection of the topic, since model validation studies tend to be tedious.

Indeed, the nature of the manuscript, being an evaluation study, makes it a bit tedious to read at a first glance. Please note that we have chosen to discuss the results both at the TOA and the surface, while discussing the individual SW and LW components of CREs as well. Furthermore, we investigate both ENSO and NAO variabilities during their two phases. While we believe that laying out such a detailed analysis might help interested readers to make further evaluations/comparisons (targeting individual piece of information), it has unavoidably

resulted into the manuscript being a bit tedious. We have tried to balance the simplicity and information content as smoothly as we possibly could.

2. One of the points that makes the article tedious to read is that the authors switch between "low resolution" and "std-res" models. It would be easier if just one qualifier is chosen, in this case probably std-res (standard resolution).

This has been corrected in the revised manuscript.

3. The quality of the figures could be better. For example, the multi-panel map plots have lots of white space between the two columns, and lots of space is occupied by large colorbars. The individual panels could be enlarged at the expense of the wasted space.

Following the reviewer suggestion, the figures are re-plotted to be clearer.

**Specific comments:**
1. page 1, line 10: no comma after whereas
2. page 1, line 18: EU-funded
3. page 4, line 23: add "the" before Niño3.4 index
4. page 26, lines 11-12: add "the" before coastal Antarctic and Arctic

All the above suggestions are incorporated in the manuscript.

---

## Author Response (AR2)

**Response to Editor's comments:**
We thank and appreciate the Editor's effort in improving the quality of the manuscript. Please find below response to your comments.

1) page 6, line 7: "opposite sign ... compensates the fluxes..."? Do you mean "opposite sign ... compensates flux errors/biases..."? Please check!
Thanks for pointing this out. The sentence is re-written as "It is evident that both in DJF and JJA averages, the opposite sign in the TOA SW and LW effects nearly compensates the biases in the fluxes over the tropics in the net effects at the TOA."

2) To my understanding you want to know whether the step from std-res to hi-res leads to improvements in CRE results. If this observation is correct, you should avoid statements like that on page 19, line 32: "... a significant improvement ... in the std-res simulations ...". What you have instead is that the transition from std-res to hi-res leads to a significant degradation in this case. There might be similar sentences at other locations in the paper. Please keep always the direction from std-res to hi-res in mind in order to make reading not even more tedious. So if the hi-res results are better, there is an improvement, but if the hi-res results are worse, there is a degradation.
Thanks for this suggestion. I understand your point. But, apart from the sentence below, the rest of the manuscript has been written in a way so as to point out if increasing the resolution has a notable impact in our results.
**Page 19, line 32:**
However, a significant improvement in the SW bias over eastern Europe in the Std-res simulations at the SFC during NAOP is seen compared to that at the TOA, whereas, the Hi-res simulations better simulate the TOA LW CREs over continental Europe.
Rephrased as "However, the Hi-res models seem to amplify the positive SW bias over Eastern Europe at the SFC during NAOP compared to the Std-res model ensemble mean. On the other hand, the Hi-res models better simulate the TOA LW CREs over continental Europe."

3) **Page 28, line 23-25:** I wonder about the speculation here. "This suggests... vary strongly among the models. ... can ... be different". This speculation is not necessary. The modellers in the author team should know whether there are differences among the models. So please reformulate this in a affirmative fashion.
This is rephrased as "This suggests that the parameterization of SW radiative transfer and the treatment of cloud optical properties vary strongly among the models."

Additionally, we would like to re-phrase the following sentence to be more precise.
**Page 13, line 4:** Although the model biases in LW at the TOA during the positive phase of ENSO are small, clear hemispherical differences can be seen at the TOA in the ENP case characterized by negative biases in the northern and positive biases in the southern hemisphere.

"Although the model biases in LW at the TOA during the positive phase of ENSO are small, clear hemispherical differences can be seen over central and eastern Pacific at the TOA in the ENP case characterized by negative biases north of 5N and positive biases south of 5N."